# SuPAR mediates viral response proteinuria by rapidly changing podocyte function

Changli Wei [1] ✉, Prasun K. Datta[2], Florian Siegerist [3,4], Jing Li[1], Sudhini Yashwanth[1], Kwi Hye Koh [5], Nicholas W. Kriho[6], Anis Ismail[7], Shengyuan Luo[1], Tracy Fischer [2], Kyle T. Amber[1,8], David Cimbaluk[6], Alan Landay[1], Nicole Endlich[3,4], Jay Rappaport[2], Michigan Medicine COVID–19 Investigators*, Salim S. Hayek [7] ✉ & Jochen Reiser [1] ✉

Elevation in soluble urokinase receptor (suPAR) and proteinuria are common signs in patients with moderate to severe coronavirus disease 2019 (COVID-19). Here we characterize a new type of proteinuria originating as part of a viral response. Inoculation of severe acute respiratory syndrome coronavirus 2 (SARS-CoV-2) causes increased suPAR levels and glomerulopathy in African green monkeys. Using an engineered mouse model with high suPAR expression, inhaled variants of SARS-CoV-2 spike S1 protein elicite proteinuria that could be blocked by either suPAR antibody or SARS-CoV-2 vaccination. In a cohort of 1991 COVID-19 patients, suPAR levels exhibit a stepwise association with proteinuria in non-Omicron, but not in Omicron infections, supporting our findings of biophysical and functional differences between variants of SARS-CoV-2 spike S1 protein and their binding to podocyte integrins. These insights are not limited to SARS-CoV-2 and define viral response proteinuria (VRP) as an innate immune mechanism and co-activation of podocyte integrins.

Viral infections are often associated with kidney barrier dysfunction, but the underlying molecular mechanisms are poorly understood[1]. Renal involvement is common in moderate to severe infection of severe acute respiratory syndrome coronavirus 2 (SARS-CoV-2)[2]. Kidney manifestations include but are not limited to proteinuria, hematuria, and tubular injury. The incidence of proteinuria in coronavirus disease 2019 (COVID-19) patients with or without concomitant acute kidney injury (AKI) ranges from 28 to 84%[1,3,4]. Both glomerular and tubular damage have been identified by autopsy of patients deceased due to COVID-19. A direct involvement of SARS-Cov-2 in kidney barrier dysfunction has not been elucidated.

Soluble urokinase receptor (suPAR), an immune-derived pathogenic factor has been implicated in glomerulopathy[5,6]. Elevated plasma suPAR levels represent innate immune activation and have been linked to podocyte integrin activation and tubular mitochondrial function[6,7]. suPAR activates αvβ3 integrin on podocytes, which in turn leads to downstream injurious signaling and proteinuria[6,8]. Notably, suPAR levels are elevated in patients with certain viral illnesses such as the European strain Puumala hantavirus (PUUV)[9], HIV-1[10,11]. Most recently, increased suPAR levels were found in patients with COVID-19, and are strongly predictive of in-hospital AKI and the need for dialysis[12]. Whether suPAR interacts with viral proteins in inducing proteinuria is unknown.

In this work, we discovered a unique functional connection between the viral protein and host suPAR, co-activating podocyte integrins and leading to a rapid proteinuric response. We showed

[1]Department of Medicine, Rush University Medical Center, Chicago, IL, USA. [2]Tulane National Primate Research Center, Covington, LA 70433, USA. [3]Department of Anatomy and Cell Biology, University Medicine Greifswald, 17487 Greifswald, Germany. [4]NIPOKA GmbH, 17489 Greifswald, Germany. [5]Morphic Therapeutic, Waltham, MA 02451, USA. [6]Department of Pathology, Rush University Medical Center, Chicago, IL, USA. [7]Division of Cardiology, Department of Internal Medicine, University of Michigan, Ann Arbor, MI, USA. [8]Department of Dermatology, Rush University Medical Center, Chicago, IL, USA. *A list of authors and their affiliations appears at the end of the paper. ✉e-mail: changli_wei@rush.edu; shayek@med.umich.edu; jochen_reiser@rush.edu

biophysical and functional differences in the spike S1 protein between Omicron and non-Omicron SARS-CoV-2 variants. These differences were particularly reflected in hospitalized COVID-19 patients, as suPAR levels showing a stepwise association with proteinuria in non-Omicron but not in Omicron infections.

## Results

### SARS-CoV-2-infected African green monkeys developed glomerulopathy

To determine whether SARS-CoV-2 infection causes glomerular barrier dysfunction, we leveraged an established nonhuman primate model of COVID-19 with African green monkeys (AGMs) (Fig. 1a)[13–15]. Approximately 24 days after inoculation of 2019-nCov when typical COVID-19-like lung pathology was developed, AGMs exhibited a rise in plasma suPAR levels (Fig. 1b) and a mild but significant increase in albuminuria (Fig. 1c). Periodic acid–Schiff (PAS) staining (Fig. 1d) and Sirius red staining (Fig. 1e) of kidney sections showed various injuries in the kidney tissues of infected AGMs, including mesangial expansion, glomerular sclerosis, glomerular proteinaceous crescents, tubular atrophy, and interstitial fibrosis. In contrast, relatively normal kidney

morphology was appreciated in the control monkeys. To examine the degree of glomerular barrier dysfunction, we performed podocyte exact morphology measurement procedure (PEMP) to analyze podocyte foot process and slit diaphragms. As compared to controls, SARS-CoV-2-infected AGMs demonstrated disorganized staining patterns of podocin and α-actinin-4 (Fig. 1f), as well as significantly reduced filtration slit density (Fig. 1g). Taken together, these data indicate that SARS-CoV-2 infection in AGMs causes elevation of plasma suPAR, proteinuria, and manifestations of glomerular disease.

### 2019-nCov spike S1 protein inoculation caused rapid proteinuria in suPAR transgenic mice

To further determine a causal role of SARS-CoV-2 infection in glomerular dysfunction, we utilized an established murine model of COVID-19 for lung pathology, which relies on inhalation of 2019-nCov spike S1 protein (Fig. 2a)[16]. Consistently, nasal inhalation of 2019-nCov S1 protein for 10 days elicited lung injuries such as septal thickening, neutrophil infiltration, edema, and focal hemorrhage in all of the three mouse strains that were tested, including wild-type C57BL/6, uPAR-depleted (Plaur⁻/⁻), and suPAR transgenic (suPAR-Tg) mice

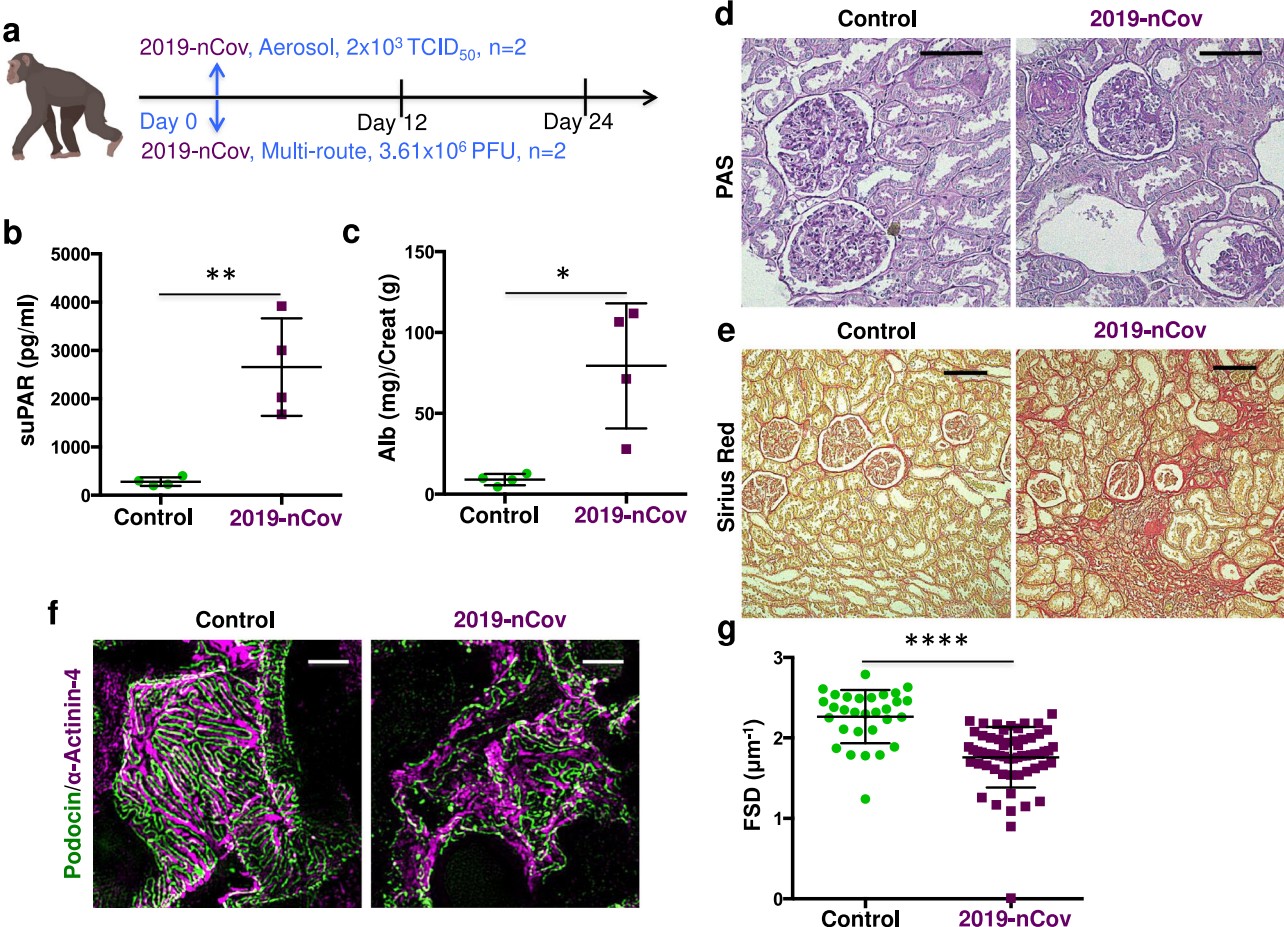

**Fig. 1 | Induction of glomerular disease in AGMs by SARS-CoV-2 2019-nCov inoculation. a** Experimental scheme for SARS-Cov-2 2019-nCov inoculation ($n = 4$ per group, 2 male, 2 female). Age and gender-matched control African green monkeys (AGMs) were either mock-infected with culture media used for virus propagation ($n = 1$) or untreated ($n = 3$ biological replicates). The illustration of AGM was created with BioRender.com. **b** Plasma soluble urokinase receptor (suPAR) levels in 2019-nCov inoculated AGMs upon nephropsy ($n = 4$ per group). $P = 0.0034$, two-tailed $t$ test. **c** Proteinuria in AGMs upon nephropsy ($n = 4$ per group). Alb, albumin; Creat, creatinine. $P = 0.011$, two-tailed $t$ test. **d** Renal histological examination, Periodic acid-Schiff (PAS) staining. Black scale bar, 100 μm.

**e** Renal histological examination, Sirius red staining. Black scale bar, 150 μm. **d**, **e** The experiments were repeated four times, generating similar results. **f** Maximum intensity projection images obtained with podocyte exact morphology measurement procedure (PEMP), indicating the disorganized slit diaphgram in "2019-nCov". White scale bar, 2 μm. **g** Calculated filtration slit density (FSD) that reflects the extent of foot process effacement ($n = 30$ replicates for control, $n = 60$ replicates for 2019-nCov). The analysis was conducted with two-sided Mann–Whitney test. **b**, **c**, **g** Data were presented as Mean ± SD. $*P < 0.05$, $**P < 0.01$, $****P < 0.0001$.

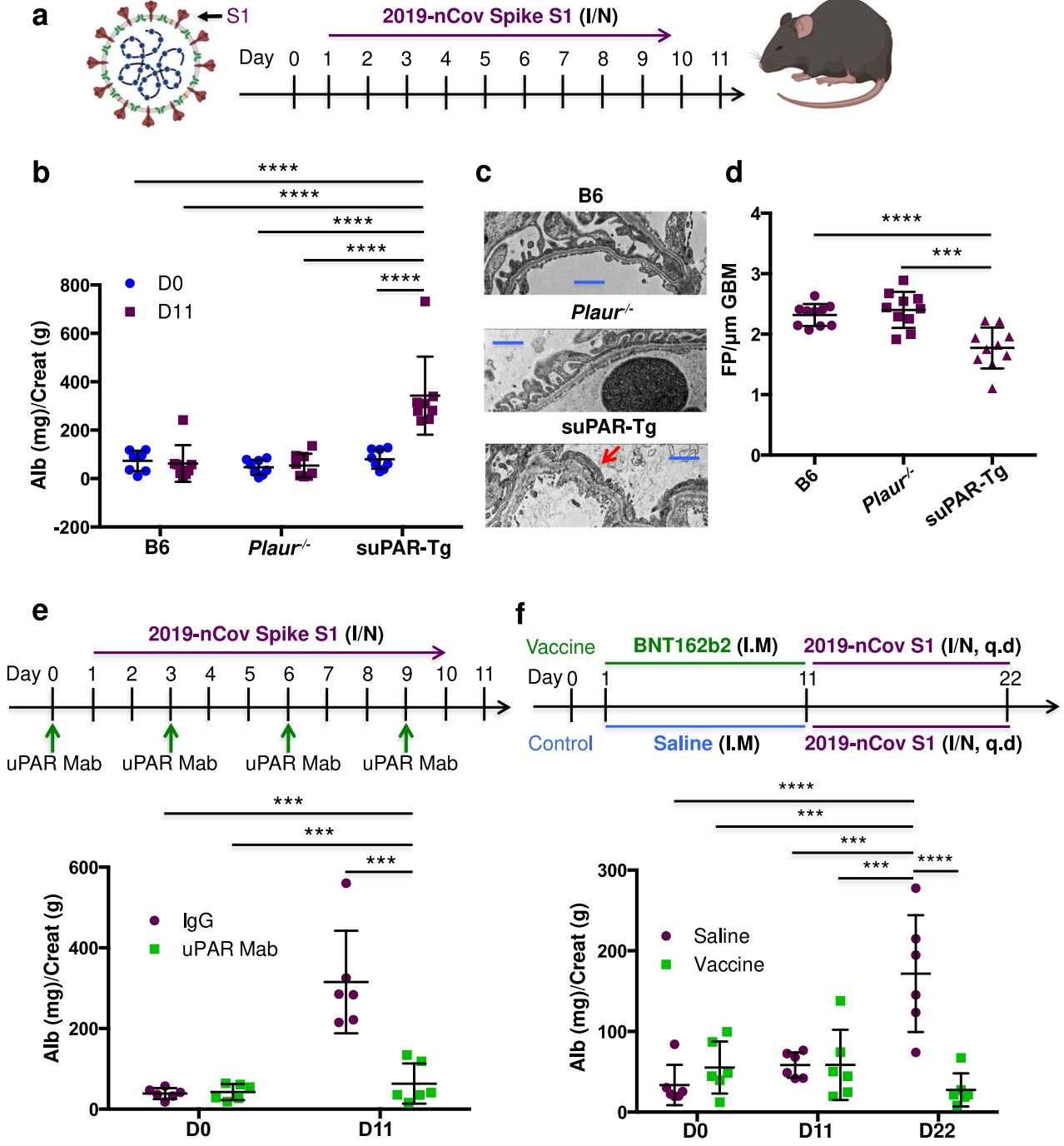

**Fig. 2 | Induction of glomerular disease in male mice with high levels of suPAR via intranasal inhalation of 2019-nCov spike S1 protein. a** Experimental scheme. I/N, intranasal (*n* = 8 mice per group, all male). The illustrations of SARS-Cov-2 and mouse were created with BioRender.com. **b** Proteinuria. 2019-nCov spike S1 protein inoculation induced proteinuria in mouse soluble urokinase receptor transgenic (suPAR-Tg) mice, but not in wild-type B6, or mouse urokinase receptor (uPAR) deficiency (*Plaur*⁻ᐟ⁻) mice (*n* = 8 per group, male). Alb albumin. Creat creatinine. **c** Transmission electron microscopy (TEM) analyses. TEM images showing glomerular filtration barrier and foot process effacement (red arrow). Representative of ten images per group. Blue scale bar, 1 μm. **d** Foot process count per μm GBM. FP foot process. GBM glomerular basement membrane (*n* = 10 replicates per group).

**e** Blocking suPAR with uPAR monoclonal antibody (Mab) as shown in treatment scheme reduced proteinuria in male suPAR-Tg mice upon 2019-nCov spike S1 protein inoculation (*n* = 6 per group). IgG isotype IgG control. D day. **f** Vaccination with single dose BNT162b2 as shown in the treatment scheme prevented 2019-nCov spike S1 protein induced proteinuria in male suPAR-Tg mice (*n* = 6 per group). Saline, phosphate-buffered saline (PBS). q.d. once a day, I.M, intramuscular. I/N, intranasal. In (**b**), (**e**) and (**f**), two-way analysis of variance with Tukey's multiple comparisons test was conducted. **d** One-way analysis of variance with Tukey's multiple comparisons test was performed. Data are presented as mean ± SD. ***$P < 0.001$, ****$P < 0.0001$.

(Supplementary Fig. 1). Interestingly, suPAR-Tg, but not wild-type C57BL/6 or uPAR-depleted mice exhibited significant proteinuria at day 11 (Fig. 2b). Transmission electron microscopy (TEM) analysis revealed podocyte foot process effacement (Fig. 2c) and a significantly

reduced filtration slit density (Fig. 2d) in 2019-nCov S1-treated suPAR-Tg mice, but not in wild-type C57BL/6 or uPAR-depleted mice.

To determine suPAR's specificity in concert with 2019-nCov S1 protein to induce proteinuria, we administered the same amount of

2019-nCov S1 protein into interleukin-5 transgenic (IL-5-Tg) mice[17]. The selection of IL-5-Tg mouse model for suPAR-Tg control is based on the following considerations: (1) it has high levels of circulating IL-5; (2) it does not have spontaneous proteinuria at baseline; (3) Like suPAR, high levels of IL-5 are often observed in severe COVID-19 patients[18]. We monitored for circulating proinflammatory cytokines and observed variable elevations of IL-2, IL-6, and TNFα in C57BL/6, suPAR-Tg, and IL-5-Tg mice after 2019-nCov S1 protein inoculation (Supplementary Fig. 2). Given this robust cytokine response in IL-5-Tg mice, we tested these animals for proteinuria. Interestingly, IL-5-Tg mice did not develop any significant proteinuria upon inoculation of 2019-nCov S1 protein (Supplementary Fig. 3). Taken together, these findings suggest a non-random synergy of S1 protein and suPAR in causing proteinuria.

## 2019-nCov S1 protein-induced proteinuria was treatable by blocking suPAR or SARS-CoV-2 vaccination

We examined if blocking suPAR could prevent 2019-nCov S1 protein inhalation-induced proteinuria. Administration of a monoclonal suPAR antibody, every 3 days for a total of four doses significantly reduced proteinuria in 2019-nCov S1 protein inoculated suPAR-Tg mice, compared to those suPAR-Tg mice received isotype IgG control (Fig. 2e). Next, we tested the effectiveness of a SARS-CoV-2 vaccine in preventing the development of proteinuria. Single intramuscular injection of Comirnaty (Pfizer-BioNTech, BNT162b2) into suPAR-Tg mice caused a constant increase of anti-spike receptor binding domain (RBD) IgG titer over monitored 22 days (Supplementary Fig. 4). Daily nasal inhalation of 2019-nCov S1 protein was initiated on day 11 after vaccination and continued for 10 days as aforementioned. As expected, suPAR-Tg mice that received vehicle control developed proteinuria with 2019-nCov S1 protein inhalation, whereas Comirnaty vaccinated suPAR-Tg mice did not (Fig. 2f). These data indicate that 2019-nCov S1 protein elicited proteinuria in suPAR-Tg mice can be prevented by either anti-suPAR antibody or SARS-CoV-2 vaccination.

## Viral protein-induced proteinuria in suPAR-Tg mice was variant and virus-specific

Next, we inoculated the same amount of nucleocapsid protein in suPAR-Tg mice to determine if other SARS-CoV-2 viral protein induces proteinuria. Mice that received nucleocapsid protein failed to develop proteinuria (Fig. 3a), even though several proinflammatory cytokines, including IFNγ, IL-1β, IL-5, and CXCL1, were elevated in blood circulation (Supplementary Fig. 5). These results indicate that while both 2019-nCov S1 protein and nucleocapsid protein elicited immune reaction, only 2019-nCov S1 protein inoculation could cause proteinuria, and it required the presence of high-plasma suPAR.

Since the initial COVID-19 breakout, there have been multiple SARS-CoV-2 variants of concern. In particular, Delta and Omicron variants have shown with different clinical manifestations. Thus we were prompted to test the spike S1 protein of Delta and Omicron variants for their respective impact on proteinuria in suPAR-Tg mice. Utilizing the same scheme as with 2019-nCov S1 protein, we inoculated two groups of suPAR-Tg mice with either Delta (B.1.617.2) S1 protein or Omicron (B.1.1.529) S1 protein. Interestingly, Delta S1 protein (Fig. 3b) but not Omicron S1 protein (Fig. 3c) inoculation induced proteinuria in suPAR-Tg mice, suggesting that SARS-CoV-2-related proteinuria is variant specific.

To examine if this viral envelop protein–host suPAR interplay involves other viruses as well, we tested the renal effect of HIV-1 gp120, Hepatitis C virus (HCV) E2 protein and Influenza B hemagglutinin A (HA) protein, respectively, in suPAR-Tg mice. As HIV-1 and HCV are non-respiratory viruses, we injected HIV-1 gp120 or HCV E2 proteins intraperitoneally into suPAR-Tg mice, whereas Influenza B HA protein was inoculated intranasally in the same scheme as with SARS-CoV-2 S1 protein. We detected proteinuria in suPAR-Tg mice that have received

HIV-1 gp120 (Fig. 3d), but not in those mice that have received HCV E2 protein (Fig. 3e) or Influenza B HA protein (Fig. 3f). Notably, injection of the same amount of HIV-1 gp120 into either wild-type C57BL/6 or $Plaur^{-/-}$ mice did not cause proteinuria (Supplementary Fig 6). These findings support the notion of specific viral protein-immune interaction in causing proteinuria.

## Variant specificity determines integrin binding affinity and potency

Next, we explored the mechanism behind SARS-CoV-2 S1 protein–host suPAR synergy in causing proteinuria. As previous studies pointed to the implication of the suPAR-integrin signaling axis in certain chronic kidney disease[5,6,19], we first examined if S1 protein interacts with suPAR. HEK293 cell-based co-immunoprecipitation analyses show that 2019-nCov spike protein but not ACE2 interacted with suPAR, while the interaction was abolished by co-treatment with two different suPAR monoclonal antibodies (Supplementary Fig. 7a). To determine protein–protein binding affinities, we performed surface plasmon resonance (SPR) assays with relevant purified proteins. Interestingly, there was no reliable binding between 2019-nCov S1 protein and suPAR with an equilibrium binding/association constant ($K_D$) > 2 mM (Supplementary Fig. 7b), suggesting that the interaction of 2019-nCov spike protein with suPAR is indirect. In contrast, 2019-nCov S1 proteins bound well to αvβ3 integrin (Fig. 4). Notably, α3β1 integrin as a binding control, did not bind S1 protein (Supplementary Fig. 8). Given SARS-CoV-2 Omicron variant differentiates itself from its predecessors in transmission and disease severity[20,21], we compared Omicron S1 protein to its 2019-nCov counterpart for ACE2 receptor and αvβ3 integrin binding affinity. 2019-nCov S1 protein bound with high affinity to ACE2 ($K_D$, 15.5 nM) (Fig. 4a) and αvβ3 integrin ($K_D$, 40.4 nM) (Fig. 4c), respectively. Whereas Omicron S1 protein bound less tightly to ACE2 ($K_D$, 96.2 nM) (Fig. 4b) and to αvβ3 integrin ($K_D$, 69.4 nM) (Fig. 4d), which may explain the differences in potency of S1 protein variants synergizing with suPAR to mediate proteinuria.

To explore the potential functional differences between the original and Omicron S1 protein for receptor binding, we utilized cultured human podocytes. 2019-nCov S1 protein treatment induced podocyte ACE2 expression (Fig. 4e) and αvβ3 integrin activity (Fig. 4f). Both effects were enhanced by the addition of suPAR into the medium. More importantly, the increases in ACE2 expression and αvβ3 integrin activity were abolished by co-treatment with LM609, a monoclonal antibody against αvβ3 integrin (Fig. 4e, f). In contrast, incubation with the same amount of Omicron S1 protein did not alter either ACE2 expression (Fig. 4e) or αvβ3 integrin activity in podocytes (Fig. 4f). While the exact molecular mechanisms remain unclear, these results indicate that 2019-nCov S1 protein is functionally distinct from its Omicron counterpart, in that 2019-nCov S1 protein can induce its receptor ACE2 expression and αvβ3 integrin activity synergizing with suPAR, while Omicron S1 protein cannot.

In addition, we observed significantly more reduction of both nephrin and podocin expression by 2019-nCov S1 protein treatment compared to its Omicron counterpart (Supplementary Fig. 9). Moreover, both S1 protein-induced podocin and nephrin reduction in podocytes were recovered by concomitant αvβ3 integrin blocking (Supplementary Fig. 9). Overall, these findings suggest that αvβ3 integrin, in addition to its known role as a binding target for suPAR, is likely also a receptor for SARS-CoV-2 spike S1 protein.

## Stepwise association of suPAR with proteinuria in patients with non-Omicron variants, but not in those with Omicron

Finally, we examined the association of circulating suPAR levels with proteinuria in 1991 patients hospitalized specifically for COVID-19 (mean age of 61.3 years, 54.0% men, 21.2% African American, 1256 infected with the non-Omicron variant and 735 with Omicron), enrolled in the Michigan Medicine COVID Cohort ($M^2C^2$), and who had

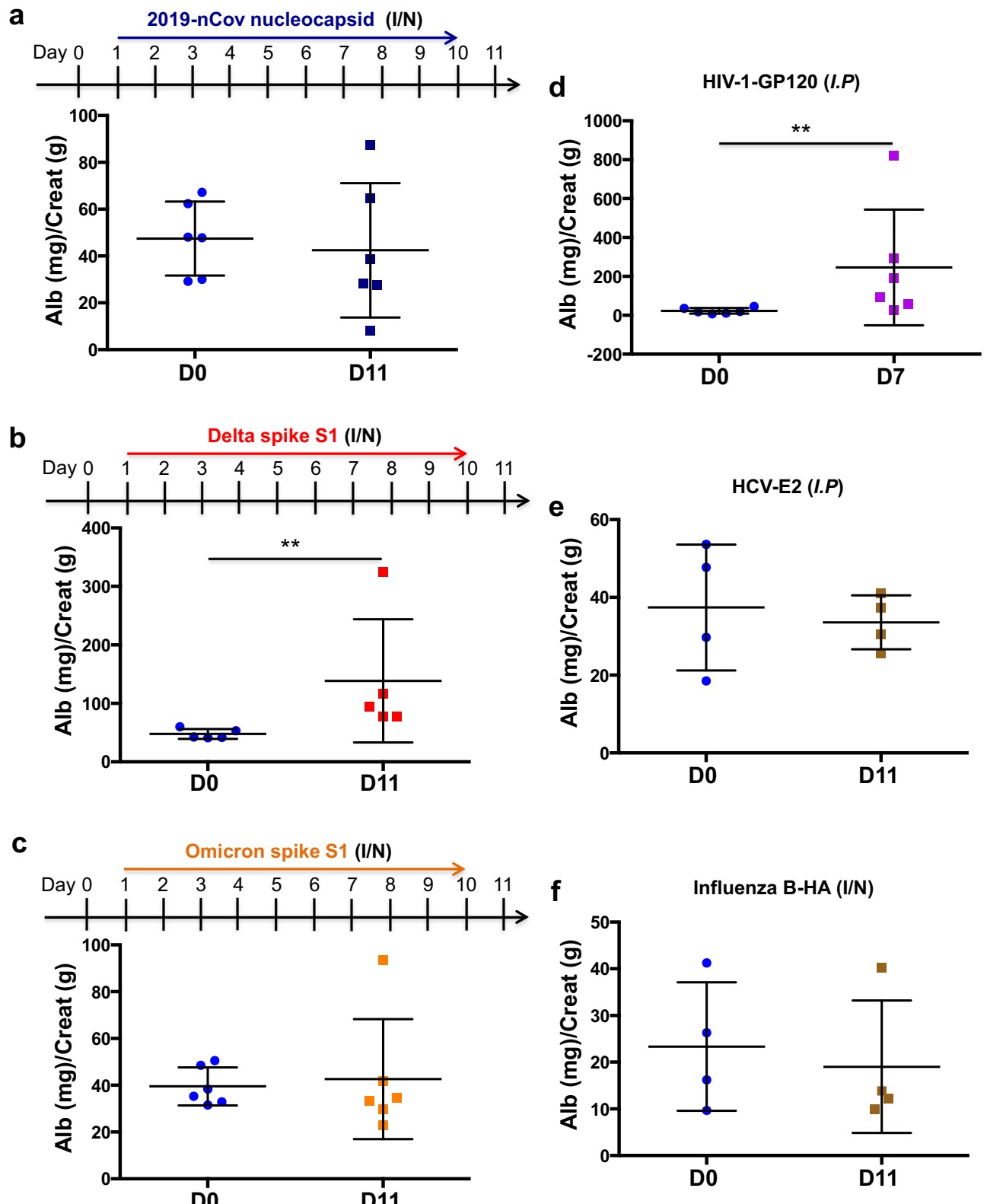

proteinuria data available (Supplementary Table 1). Patients hospitalized due to Omicron within 48 h of admission were less likely to have ≥1+ proteinuria compared to those with non-Omicron infections (54.6% and 64.2%, respectively, $P < 0.001$). Similarly, suPAR levels were lower in patients hospitalized with Omicron compared to non-Omicron variants (median of 5.67 ng/ml [IQR 4.07–8.18] vs. 8.00 ng/ml [IQR 5.46–12.3], $P < 0.001$). These associations were

independent of clinical characteristics, including age, gender, race, serum creatinine, diabetes mellitus, hypertension, heart failure and coronary artery disease, admission eGFR, and respiratory failure requiring mechanical ventilation (Supplementary Tables 2 and 3).

Most importantly, we found that the association between suPAR levels and proteinuria differed significantly according to the predominant variant: suPAR levels were associated with proteinuria in

**Fig. 3 | Viral protein-caused proteinuria in suPAR-Tg mice was SARS-CoV-2 variant-dependent and virus-specific. a** Inoculation of SARS-CoV-2 2019-nCov nucleocapsid protein in the same scheme as with 2019-nCov S1 protein did not cause proteinuria in male mice with high levels of suPAR ($n = 6$). $P = 0.474$, non-significant, two-sided Mann–Whitney test. **b** Inoculation of SARS-CoV-2 Delta variant (B.1.617.2) S1 protein induced proteinuria in male mice with high levels of suPAR ($n = 5$). $P = 0.0079$, two-sided Mann–Whitney test. **c** Inoculation of SARS-CoV-2 Omicron variant (B.1.1.529) S1 protein in the same scheme as with 2019-nCov S1 failed to elicit proteinuria in male suPAR-Tg mice ($n = 6$ biological replicates).

$P = 0.5714$, non-significant, two-sided Mann–Whitney test. **d** Injection of human immunodeficiency virus type 1 (HIV-1) envelop gp120 protein caused proteinuria in suPAR-Tg mice ($n = 6$, 3 male, 3 female). $P = 0.0087$, two-sided Mann–Whitney test. **e** Injection of hepatitis C virus (HCV) E2 protein did not cause proteinuria in suPAR-Tg mice ($n = 4$, 2 male, 2 female). $P = 0.8286$, non-significant. **f** suPAR-Tg mice inoculated with influenza B HA protein did not develop proteinuria ($n = 4$, male). HA hemagglutinin A. $P = 0.6571$, non-significant, two-sided Mann–Whitney test. I/N intranasal. I.P intraperitoneal. Data are presented as mean ± SD in all graphs. **$P < 0.01$.

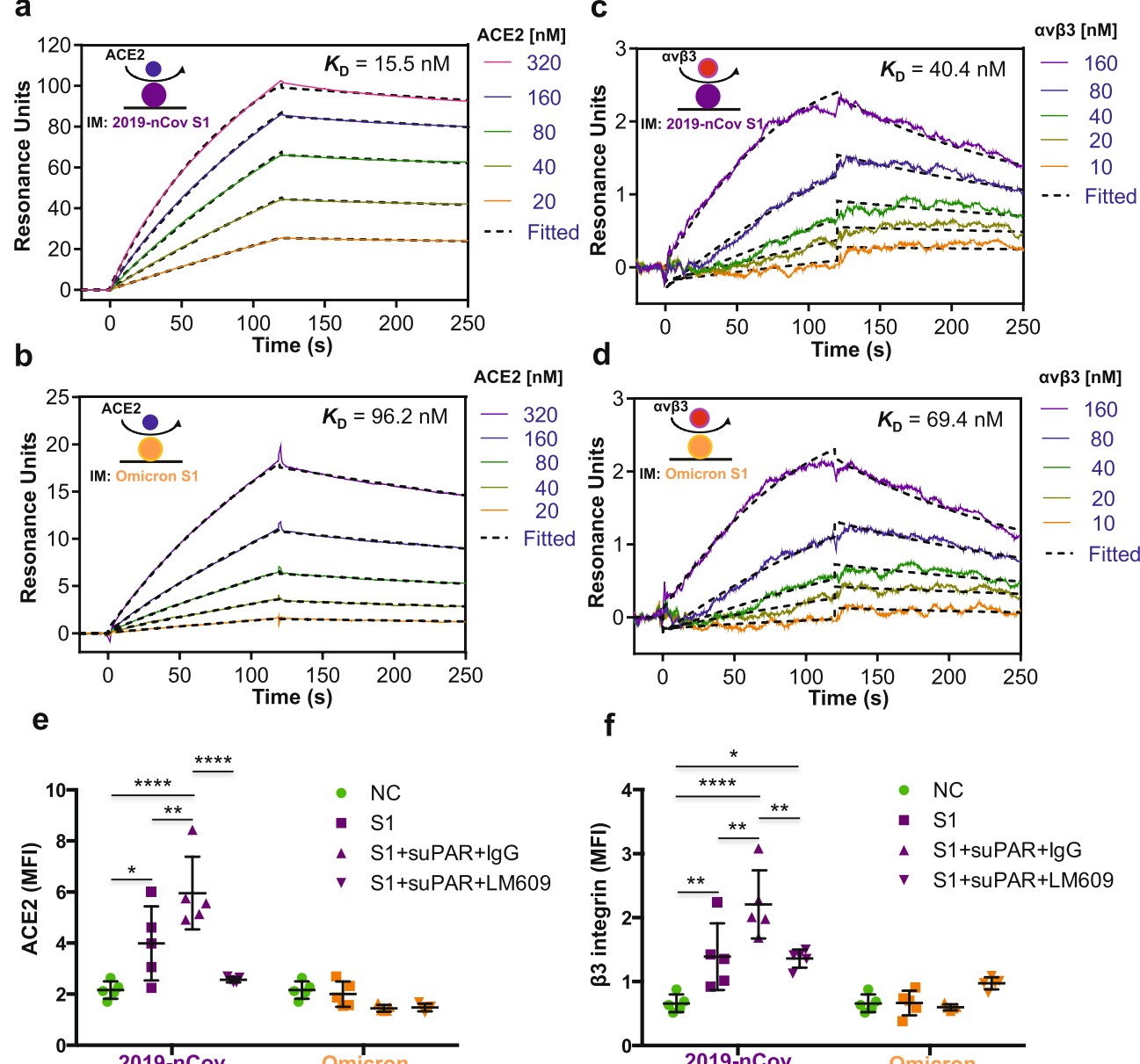

**Fig. 4 | Omicron S1 protein distinguished from its 2019-nCov counterpart both biophysically and functionally. a–d** S1 protein binding affinity as indicated by surface plasmon resonance assays. As shown, S1 protein was immobilized onto a CM5 sensor chip, while αvβ3 integrin or ACE2 was applied as an analyte in a series of increasing concentrations. Calculation of $K_D$ value indicates that 2019-nCov S1 protein bound more tightly to ACE2 (**a**) and αvβ3 integrin (**c**), as compared to Omicron S1 protein (**b**, **d**). **e**, **f** The effects of S1 protein on cultured human podocytes. Fully differentiated human podocytes were treated for 16 h before harvest for

immunofluorescence staining and imaging ($n = 5$ biological replicates). 2019-nCov but not Omicron S1 protein enhanced ACE2 expression in human podocytes (**e**). 2019-nCov but not Omicron S1 protein induced αvβ3 integrin activity in human podocytes, as indicated by AP5 (an antibody specific for αvβ3 integrin activation) immunostaining (**f**). Notably, $P < 0.01$ for 2019-nCov S1 compared to Omicron S1 in both (**e**, **f**). MFI mean fluorescence intensity. Data are presented as mean ± SD. The analyses were conducted with two-way analysis of variance with Tukey's multiple comparisons test. *$P < 0.05$, **$P < 0.01$, ***$P < 0.001$, ****$P < 0.0001$.

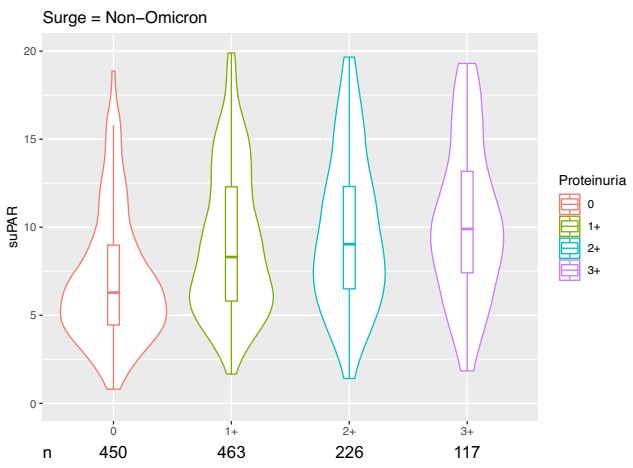
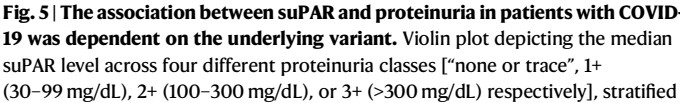
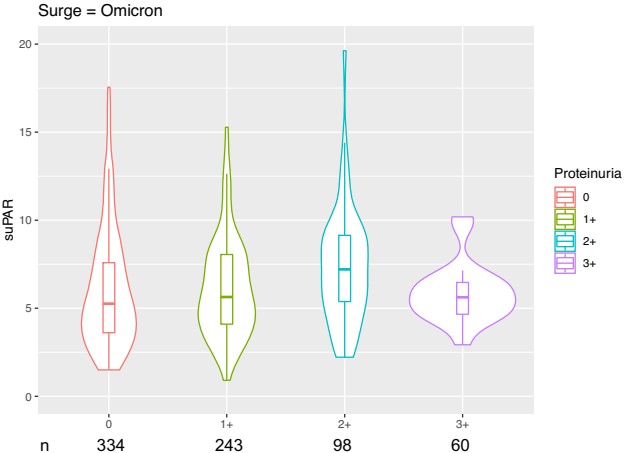

**Fig. 5 | The association between suPAR and proteinuria in patients with COVID-19 was dependent on the underlying variant.** Violin plot depicting the median suPAR level across four different proteinuria classes ["none or trace", 1+ (30–99 mg/dL), 2+ (100–300 mg/dL), or 3+ (>300 mg/dL) respectively], stratified by non-Omicron and Omicron SARS-Cov-2 infections. The central line in box represents median, box represents the interquartile, and whiskers represent the minimum and maximum, respectively. Plot was generated using R with ggplot2 package. The sample size (*n*) of each class is shown below the graphs.

patients infected with the non-Omicron variant (odds ratio, OR 2.07, 95% CI [1.75–2.45]), but not in patients infected with Omicron (OR 1.35, 95% CI [0.97–1.88]) (*P* = 0.023 for interaction). Patients infected with non-Omicron variants exhibited a stepwise increase in suPAR levels with increasing severity of proteinuria, but this association was heavily attenuated in patients infected with the Omicron variant (Fig. 5).

To determine the nature of the proteinuria as glomerular or tubular in patients with COVID-19, we performed urine electrophoresis in a randomly selected subset of patients from $M^2C^2$ with and without AKI. Interestingly, our results revealed an albumin-predominate urine protein profile whether the patients had AKI or not, suggesting that the proteinuria in COVID-19 is predominantly of glomerular origin (Supplementary Fig. 10). Overall, these data align with the aforementioned functional and biophysical results, implying that suPAR associates with SARS-CoV-2 spike S1 protein to cause proteinuria in a variant-dependent manner.

## Discussion

In this study, we present experimental and clinical evidence indicating a unique interaction between viral proteins and suPAR as well as αvβ3 integrin in causing glomerular barrier dysfunction, adding viral infection as a possible cause for proteinuria. We demonstrate a differential association between suPAR levels and proteinuria in patients hospitalized for COVID-19 that depends on the underlying predominant viral strain. These clinical observations were corroborated nicely in murine models of COVID-19, in which we showed that exposures to the original 2019-nCov spike S1 protein, its Delta variant but not Omicron counterpart, are potent inducers of proteinuria. This pathological effect was dependent on high suPAR levels, likely through further enhancement of podocyte integrin activation and slit diaphragm dysfunction, which can be abrogated by anti-suPAR antibody in vivo and anti-αvβ3 integrin blockade in vitro. Overall, these findings provide specific insights into the mechanisms of COVID-19-associated glomerular dysfunction. Given that other virus (i.e., HIV-1) can also cause rapid proteinuria in suPAR-Tg mice, our findings may have broad relevance to the interaction between viral proteins and immune mediators in virus-associated proteinuric diseases. While suPAR is known to be a strong risk factor for kidney diseases, including focal segmental glomerulosclerosis (FSGS)[5,6,8,11], its participation in orchestrating viral response proteinuria (VRP) bears the question whether infections play a role in the rapid onset and relapse of FSGS including recurrent FSGS after transplantation.

Exactly why non-Omicron SARS-Cov-2 S1, HIV-1 gp120, but not other tested viral proteins including HCV E2 and Influenza B HA elicited proteinuria in mice with high levels of suPAR needs further study. Both HIV infection and COVID-19 are associated with a wide range of renal manifestations, including collapsing focal segmental glomerulosclerosis[22]. Both direct and indirect renal cytotoxic effects could be implicated in these scenarios[23]. At molecule level, suPAR avidly binds to and activates αvβ3 integrin leading to proteinuric kidney disease[6]. Integrins are entry receptors for numerous viruses associated with proteinuric kidney disease[24]. For example, α4β7 has been implicated in HIV-1 cell attachment[25]; while αvβ3 and αvβ5 integrin are involved in HIV-1 transactivating factor (Tat)'s interaction with cell surface[26,27]. α5β1 and αvβ6 are associated with Epstein-Barr virus (EBV) cell entry[28,29]; whereas αvβ3 and αIIbβ3 render cells permissive to PUUV[30]. Interestingly, PUUV exhibits similar renal cell tropism as SARS-CoV-2, with infections commonly associated with proteinuria[31]. We have described podocyte damage in patients with PUUV, with the severity of proteinuria positively correlating with suPAR levels[9]. In contrast, it is not yet clear if any integrin is required in the process of either HCV or Influenza virus infection. Here, our study implies that increased suPAR levels and certain viral infections could cause proteinuria in a synergistic manner. Given the association of viral infections with proteinuria syndromes, our discovered mechanism of virus-associated proteinuria suggests that proteinuria can be infectious in etiology and potentially be treated with anti-virals or through blockade of the suPAR-integrin cascade.

So far, direct kidney infection by SARS-CoV-2 remains controversial, as proof of direct infection came largely from autopsy kidney tissues but was rarely seen in biopsy tissues[32–35]. We explored if kidney cells require direct infection by SARS-Cov-2 or if viral proteins can be sufficient in causing proteinuria. Multiple viral proteins have been detected in blood circulation, due to direct shedding, exosome formation or defective proviruses[36–40]. The presence of viral proteins in blood circulation or local cellular niche could represent an alternative path for a virus to harm host cells, as viral proteins could engage local host cells even without the presence of viruses for direct infection. Notably, in-situ HIV viral replication is not indispensable, as the expression of some HIV viral genes was sufficient to generate HIV-associated nephropathy (HIVAN)-like pathology in rodents[41,42]. With COVID-19, integrins with arginyl-glycyl-aspartic acid (RGD) domain, in addition to ACE2 have recently been shown to facilitate SARS-CoV-2 cell entry[43]. SuPAR's target receptor in podocytes, αvβ3 integrin binds

strongly to SARS-CoV-2[44]. Here, we show that the 2019-nCov spike S1 protein induces αvβ3 integrin activity on podocytes, which, in the presence of high suPAR, leads to reduced nephrin and podocin expression. αvβ3 integrin is located on the podocyte membrane and COVID-19-infected podocytes may shed viral protein to activate integrin in an autocrine or paracrine fashion. Alternatively, our data suggest that viral cell entry is not a prerequisite for glomerular dysfunction, and that shedded viral proteins alone can induce proteinuria in the presence of high suPAR levels.

The $M^2C^2$ is unique in that it prospectively enrolled patients hospitalized specifically for COVID-19 (excluding those infected with SARS-CoV-2 and hospitalized for other reasons), in whom suPAR levels were measured from blood samples collected on admission−minimizing the confounding effects of treatment. The limitations of the cohort include the semi-quantitative measurement of proteinuria through urinalysis, and the assumption of predominant SARS-CoV-2 variant exposure based on variant of concern time cutoff−limitations we believe are overcome by the relatively large study population.

In conclusion, our findings provide a biological basis for the distinctive glomerular effects of different variants of SARS-CoV-2 as observed in COVID-19 patients and experimental models. It may serve as a mechanistic framework for viral-immune molecular interactions, explaining an infection-associated proteinuric response that ultimately may progress to glomerular diseases.

## Methods

### Nonhuman primate SARS-CoV-2 infection model

The Institutional Animal Care and Use Committee of Tulane University reviewed and approved [Nonhuman Primate (NHP)] Model Development for SARS-CoV-2 Infection, #P0435) and all the procedures for this study. All animals were cared for following the ILAR Guide for the Care and Use of Laboratory Animals 8th Edition. The Tulane University Institutional Biosafety Committee approved sample handling, inactivation, and transfer procedures from BSL3 containment.

We used AGMs of the *Chlorocebus aethiops* species (n = 4, two males and two females, 16 years old) to model SARS-CoV-2 infection in nonhuman primates for up to 4 weeks. The AGMs were wild caught in St. Kitts and Nevis and supplied by a USA importer under permit number 524/2019 dated 22/01/2019 (CITES Management Authority, Ministry of Agriculture, St. Kitts and Nevis) and U.S Fish and Wildlife Services (Import License number 110542 dated 01/29/2019, and housed at Tulane National Primate Research Center, Covington, for this study. The AGMs were inoculated with laboratory-grown 2019-nCoV/USA-WA1/2020 strain of SARS-CoV-2 (MN985325.1) as described previously for a case report[45]. Two animals exposed to aerosol received an inhaled dose of -2 × 10³ 50% tissue culture infectious dose ($TCID_{50}$). The other two animals (AGM2 and AGM3) were exposed through multiple routes (oral, 1 mL; nasal, 1 mL; intratracheal, 1 mL; and conjunctival, 50 μL per eye) resulting in a cumulative dose of $3.61 × 10^6$ plaque-forming units as reported previously for SARS-Cov-2 induced acute respiratory distress in aged AGMs[13]. Matched control AGMs were either mock-infected through multiple routes with tissue culture media (n = 1) or nontreated (n = 3) as reported previously for SARS-Cov-2-associated neuropathology in nonhuman primates[46].

All AGMs were housed for over a year before assignment to the study and tested negative for simian type D retrovirus, simian immunodeficiency virus, simian T-cell lymphotropic virus type 1, measles virus, macacine herpesvirus 1, and tuberculosis. The monkeys were assessed twice daily. Pre-exposure and postexposure samples were collected. Physical examination, plethysmography, and imaging (radiography and positron emission tomography/computed tomography) were performed 7 days before exposure and then weekly thereafter[46]. Animals were euthanized for necropsy 24−28 days post exposure.

The following endpoints were assessed: plasma suPAR level, proteinuria, kidney histopathology, and glomerular slit diaphragm size as measured using podocyte exact morphology measurement procedure (PEMP).

### PEMP analysis

Immunofluorescence staining for PEMP was performed as previously described for nephrin and DAPI with minor modifications[47]. In brief, after deparaffinization and rehydration, kidney sections (2 μm) were boiled in Tris-EDTA buffer (10 mM Tris, 1 mM EDTA, pH 9) in a pressure cooker for 5 min. The following primary antibodies were incubated overnight at 4 °C: rabbit anti-podocin (IBL International, JP29040, 1:100), mouse anti-α-actinin-4 (Santa Cruz, sc-390205, 1:50) followed by a blocking step for 20 min. After washing in PBS, secondary antibodies were incubated for 1 h at room temperature. DAPI (1:100) was then added to the slides for 5 min, followed by PBS wash. For 3D SIM, z-stacks of 21 planes of both channels (488 and 561 nm) were acquired using the N-SIM-E super-resolution microscope (Nikon) equipped with a ×100 silicone objective. The images were reconstructed into 3D SIM images using NIS-Elements AR 5.30 (Nikon). The 3D images were stitched with NIS-Elements Ar 5.21.02. The z-stacks were converted into a maximum intensity projection (MIP) followed by the automatized identification of the filtration slit length. The filtration slit density, i.e., the length of the filtration slit per podocyte foot process area, was determined. Filtration slit density of minimum five entire glomerular cross sections was determined for every sample.

### SARS-CoV-2 spike S1 protein inoculation mouse models

All mouse work in this study was approved and overseen by Rush University Institutional Animal Care and Use Committee under protocol #22-019. All activities were conducted in compliance with the Animal Welfare Act (AWA), other federal statutes and regulations, and state laws and directives related to animals. Mouse strains: wild-type C57BL/6j mice (000664), uPAR knockout (*Plaur*⁻/⁻) mice (002829) were purchased from the Jackson Laboratory; suPAR transgenic (suPAR-Tg) mice that carry high level of circulating mouse suPAR driven by AP2 promoter were developed in house and characterized before[5,8]. CD2-driven interleukin-5 transgenic (IL-5-Tg) mice[17] were provided by Dr. Kyle Amber. All mice were housed and bred in the animal care facility under a 12/12-h light/dark cycle, 20 °C ambient temperature with an average of 45% humidity.

To establish the SARS-CoV-2 spike protein inoculation model, 10−12-week-old male C57BL/6j, *Plaur*⁻/⁻, and suPAR-Tg mice (n = 8 in each group) were inoculated intranasally (I/N) with recombinant 2019-nCov spike S1 (Abeomics, 32-190005) at 2.5 ng/g body weight, once a day for 10 days. In brief, the S1 protein was diluted in normal saline with a final concentration of 25 ng/μL. Mice were held in a supine position and 1 μL of diluted S1 protein was delivered into each nostril using a Pipetman. Control mice received 2 μl of saline instead. To evaluate the specificity of high level of suPAR in this SARS-CoV-2 spike protein inoculation model for proteinuria development, the same amount of 2019-nCov S1 protein was inoculated into age and gender-matched IL-5-Tg mice (n = 3).

To test the effect of suPAR blocking, age and weight-matched male suPAR-Tg mice were randomly divided into two groups. The treatment group (n = 6) received rat-anti-mouse uPAR monoclonal antibody (R&D Systems, MAB531) at a dose of 500 μg/kg body weight intraperitoneally (i.p.), while the control group (n = 6) received the same amount of rat IgG2a isotype control (R&D Systems, MAB006) via the same route. uPAR antibody or IgG2a control was given every 3 days starting at day 0 for four times.

To test the effect of SARS-CoV-2 vaccine, age and weight-matched male suPAR-Tg mice were randomly divided into two groups (n = 6 in each group). Vaccine group received a single dose of BNT162b2

(Comirnaty, Pfizer-BioNTech) at 0.25 μg/g body weight (i.m.). Control mice received the same amount of phosphate-buffered saline (PBS) via the same route. Eleven days later, all these 12 mice were inoculated with 2019-nCov S1 protein as aforementioned. In particular, peripheral blood was drawn before vaccination (day 0), on day 7, 11, 22 post vaccination to monitor the titer of IgG antibody to spike RBD protein with GENLISA mouse anti-SARS-CoV-2 (COVID-19) IgG antibody to spike RBD protein Quantitative TITRATION ELISA kit (Krishgen Biosystems).

To evaluate the effects of SARS-CoV-2 nucleocapsid protein (Abeomics, 21-1003), Delta (B.1.617.2, Sino Biological, 40591-V08H23) or Omicron (B.1.1.529, Sino Biological, 40591-V08H41) variant spike S1 protein, the same amount of the recombinant protein was inoculated into suPAR-Tg mice ($n = 6$ for nucleocapsid protein, Omicron S1 protein, respectively; $n = 5$ for Delta S1 protein) using the same scheme as with the original 2019-nCov spike S1 protein. Blood and urine were collected before the 1st dose (day 0) and after the last dose (day 11) for further analyses.

The following endpoints were assessed: kidney function as measured by urine albumin, serum creatinine, and blood urea nitrogen, serum levels of spike S1 protein, serum inflammatory cytokines, and kidney histology and glomerular ultrastructural analyses.

### Non-SARS-CoV-2 viral protein mouse models

To determine the potential effects of non-SARS-CoV-2 viral proteins on proteinuria development, we utilized 10–12-week-old suPAR-Tg mice that have high levels of circulating suPAR. For Influenza (B/Florida/4/2006) hemagglutinin (HA) protein (eEnzyme, 1A-B2-005P), 2.5 ng/g body weight in saline was given daily intranasally for 10 days ($n = 4$, male). For HCV E2 protein (subtype 1a, eEnzyme, HCV-E2-015P), 5 ng/g body weight in saline was given daily intraperitoneally for 10 days ($n = 4$, 2 male, 2 female). For HIV-1/Clade B gp120 protein (eEnzyme, IV-RB1-005P), 5 ng/g body weight in saline was given daily intraperitoneally for 7 days ($n = 6$, 3 male, 3 female). Urine was collected before and after viral protein treatment for analysis. To test the effects of HIV-1 gp120 in wild-type C57BL/6 mice ($n = 10$, 5 male, 5 female) and *Plaur*$^{-/-}$ mice ($n = 10$, 5 male, 5 female), age and gender-matched mice were administered the same amount of HIV-1 gp120 intraperitoneally for 7 days.

### Mouse sample processing and analyses

Urine albumin was measured with mouse albumin ELISA kit (Bethyl, E99-134) and creatinine was measured with a creatinine colorimetric assay kit (Cayman, #500701). Serum creatinine and BUN were assayed with creatinine assay kit (Sigma, MAK080) and urea nitrogen colorimetric detection kit (Invitrogen, EIABUN). Serum levels of 2019-nCov spike S1 protein and Omicron were measured by SARS-CoV-2 (2019-nCov) spike ELISA kit and SARS-CoV-2 Omicron (B.1.1.529) variant spike ELISA kit (Sino Biological), respectively, following the manufacturer's instructions. TEM analysis with mouse kidney tissues was performed as reported previously[8]. Circulating proinflammatory cytokines, including IFNγ, IL-1β, IL-2, IL-4, IL-5, IL-6, KC/GRO, IL-10, IL-12p70, and TNFα were measured with multi-spot assay system, proinflammatory panel 1 (mouse) kits (MSD).

### Immunoprecipitation assay

We transfected HEK-293T cells with plasmid DNAs encoding, respectively, human suPAR1 (C-terminal Myc/Flag tag), ACE2 (C-terminal C9 tag), SARS-CoV-2 spike (C-terminal C9 tag) and integrin aV subunit (C-terminal V5-HIS tag) fusion proteins using Fugene 6 reagent (Promega, E2692). Thirty-six hours after transfection, cells were harvested and cell pellets were resuspended in cell lysis buffer with Protease inhibitor cocktail (Roche, 04693116001) and Phosphatase Inhibitor Cocktail 2 & 3 (Sigma, P5726 and P0044). Co-immunoprecipitation was performed using a Pierce Protein A/G Agarose (Thermo Scientific, 20421) or according to the manufacturer's instructions. Briefly, the Myc mouse

monoclonal antibody (Sigma, M4439), mouse anti-V5 antibody (Invitrogen, 46-075) was added to the precleared cell lysate, respectively and was rotated overnight at 4 °C, then, incubated with A/G Agarose slurry for 4 h. Samples were analyzed by immunoblotting with anti-C9 antibody (Abnova, PAB26959, 1:1000) to detect spike protein or ACE2, while mouse anti-Flag antibody (Sigma, F1804, 1:1000) was applied to detect suPAR1. The presence of integrin aV subunit was detected by monoclonal anti-His antibody (Invitrogen, MA1-21315, 1:1000). Experiments were repeated three times.

### Biacore surface plasmon resonance assay

Protein interactions were measured and analyzed on a Biacore T200 instrument (GE Healthcare) as described previously for determining the binding affinities between APOL1 and αvβ3 integrin with minor modifications[19]. In brief, SARS-CoV-2 (2019-nCov) Spike S1 (Abclonal, RP01262), SARS-CoV-2 Omicron (B.1.1.529) Spike S1 (Sino Biological, 40591-V08H41) was immobilized to flow channels on a CM5 sensor chip using a standard amine-coupling method. Human integrin αvβ3 (R&D Systems, 3817-AV-050) or human ACE2 (Abclonal, RP01266) with a series of increasing concentrations (0–160 nM at twofold dilution) as an analyte was applied to all tested channels at a 10 μL/min flow rate at 25 °C. Using the Biacore T200 evaluation software 2.0.3, sensorgrams were analyzed, and response units (RU) were measured during the equilibration phase at each concentration for steady-state affinity fittings. Data were referenced with blank (enthanolamine) RU values on flow channel 1. Kinetic fittings were done by 1 to 1 Langmuir binding model. Experiments were repeated twice.

### Human podocyte culture and treatment

We cultured the conditionally immortalized human podocytes as described previously[48]. The use of this cell line was under Rush IBC approval # 22051103-IBC01. In brief, podocytes were proliferated and maintained at 33 °C in RPMI-1640 medium (Invitrogen), containing 10% FBS and 1% of insulin–transferrin–selenium (Sigma). Cultured podocytes were seeded on coverslips and allowed for differentiation for 14 days at the growth-restrictive temperature of 37 °C before any treatment. After 14 days' culture at 37 °C, human podocytes were treated for 16 h with 2019-nCov spike S1 or its Omicron counterpart (1 ng/mL), together with human suPAR (R&D Systems, 807-UK-100/CF, 10 ng/mL), an αvβ3 integrin monoclonal antibody (Millipore, MAB1976, clone LM609) or mouse IgG1 isotype control (Invitrogen, 02-6100) at 500 ng/mL. After fixation and blocking, the cells were incubated with the primary antibodies, including goat anti-human ACE2 (R&D Systems, AF933, 1:100), mouse anti-human β3 integrin antibody AP5 (Kerafast, EBW107, Clone AP5, 1:50), mouse anti-human nephrin (a gift from Dr. Karl Tryggvason, 1:50) and rabbit anti-human podocin (Sigma, P0372, 1:150), respectively, followed by Alexa Fluor-conjugated chicken anti-mouse IgG (Introgen, A21200, 1:1500), chicken anti-rabbit IgG (Introgen, A21442, 1:1500) or chicken anti-goat IgG (Invitrogen, A21468, 1:1500). The images were acquired using a LSM 700 confocal microscope (Carl Zeiss). For each treatment, five ×20 images were taken from each of the four quadrants and the center. The mean fluorescence intensity (MFI) for target proteins (red or green channel) from each section was corrected with that of DAPI. Experiments were repeated three times.

### SuPAR levels and proteinuria in patients with COVID-19

The Michigan Medicine COVID-19 Cohort (M$^2$C$^2$) study was approved by the University of Michigan's Human Research Protection Program (HRPP) and the Institutional Review Boards of the University of Michigan Medical School (IRBMED). M$^2$C$^2$ is an ongoing prospective observational study and biobank which systematically enrolled adult patients hospitalized specifically for COVID-19 since the onset of the pandemic (March 1, 2020).

Details of the cohort have been published elsewhere[49]. The work contained within this study is covered under protocol number HUM00178971. A waiver of informed consent was obtained, and there was no compensation for participation for this study. SuPAR levels were measured in serum samples collected from hospitalized patients within 48 h of admission using the suPAR-nostic assay (Virogates, Copenhagen, Denmark). Proteinuria data was derived from dip-stick urinalysis and classified as "none or trace", 1+ (30–99 mg/dL), 2+ (100–300 mg/dL), or 3+ (>300 mg/dL). SARS-CoV-2 infections prior to 10/01/2021 were classified as secondary to the non-Omicron variants, while those occurring between 10/01/2021 and 11/01/2022 were classified as secondary to the Omicron variant. The cutoff of 10/01/2021 was derived from the State of Michigan COVID-19 case reporting website (https://www.michigan.gov/coronavirus/stats) as the initial surge in Omicron cases in Michigan.

We compared the median suPAR level and the incidence of proteinuria between non-Omicron and Omicron hospitalizations using the Mann–Whitney $U$ test and Chi-square test, respectively, for univariable analysis. In multivariable analysis, we first used linear regression to assess whether differences in suPAR levels (dependent variable) between predominant SARS-CoV-2 variant (independent variable) were independent of clinical characteristics and disease severity, adjusting for age, gender, race, hypertension, diabetes mellitus, coronary artery disease, heart failure, admission estimated glomerular filtration rate (eGFR) and whether the patient required invasive mechanical ventilation. We then used binary logistic regression to assess the association between variant (non-Omicron vs. Omicron, independent variable) and proteinuria (at least 1+ vs. none or trace, dependent variable) adjusting for the aforementioned characteristics. To determine whether the association between suPAR levels and proteinuria was dependent on the predominant SARS-CoV-2 variant, we examined the interaction term suPAR*variant in a binary logistic regression model with proteinuria as the dependent variable. Lastly, we plotted the median suPAR levels stratified by severity of proteinuria and predominant SARS-CoV-2 variant using R with ggplot2 package.

### Human urine electrophoresis (UEP)
We randomly selected 20 patients with at least 1+ proteinuria (10 with, 10 without AKI) from the above M²C² cohort and run UEP to determine the urinary protein profiling.

Frozen human urine samples collected from M²C² cohort were thawed on ice and centrifuged at 10,000 rpm for 1 min with precipitate removed. Total urinary protein concentration was determined by Pierce™ Coomassie Plus (Bradford) Assay Kit (ThermoFisher Scientific, 23236). An equal amount of urine in total protein from each patient was loaded into NuPage 4–12% Bis-Tris gels (The ThermoFisher Scientific, NP0322BOX) for electrophoresis at 200-volt for 35 min. Bovine serum albumin (BSA) (Biorad, 500-0207, 2 µg) was loaded as a positive control for albumin. After electrophoresis, the protein gels were carefully removed from the cast, washed briefly with ddH₂O, then stained with Imperial protein stain (ThermoFisher Scientific, 24615) for 1 h at room temperature. After sufficient washing with ddH₂O to remove background, images were taken with a digital camera for analysis.

### Statistics
Results are presented as mean ± SD unless stated otherwise. Comparisons between two groups were performed with either Student's $t$ test or Mann–Whitney test where appropriate. Comparisons between multiple groups were conducted with one-way ANOVA or two-way ANOVA with Tukey's multiple comparison test. A two-tailed $P$ value of less than 0.05 was considered significant. GraphPad Prism 6 was used to perform statistical analysis and to generate figures for nonhuman subjects. Analyses of COVID-19 patient data were performed in SPSS Statistics 27.

### Reporting summary
Further information on research design is available in the Nature Portfolio Reporting Summary linked to this article.

## Data availability
Source data are provided with this paper.

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

## Acknowledgements

We thank Drs. Steve Mangos, Varsha Suresh Kumar, Mehmet M Altintas, Sarfaraz Ahmad from Rush University; Tim Endlich, Vedran Drenic, and Nihal Telli from Nipoka for their technical support. We thank Dr. Kalipada Pahan from Rush University for his insights and support on spike S1 inhalation modeling. Nonhuman primate studies were supported by P51OD011104 and Fast Grant to T.F.

## Author contributions

C.W. and J. Reiser conceptualized the project. C.W., S.S.H., and J. Reiser analyzed the data and wrote the paper. P.K.D., T.F., and J. Rappaport performed AGM COVID-19 modeling and measurements. F.S. and N.E. performed PEMP analyses. C.W., J.L., S.Y., K.H.K., N.W.K., S.L., and D.C. performed mouse modeling, biochemical experiments, and histological examinations. K.T.A. provided IL-5 transgenic mice. A.L. helped coordinate the study. A.I. performed urine analyses with patient samples. S.S.H. obtained and analyzed the patient data.

## Competing interests

J. Reiser is co-founder and shareholder of Walden Biosciences, a biotechnology company that develops novel kidney protective therapies. The remaining authors declare no competing interests.

## Additional information

## Michigan Medicine COVID−19 Investigators

Alexi Vasbinder[7], Anis Ismail[7], Elizabeth Anderson[7], Tonimarie Catalan[7], Ian Pizzo[7], Brayden Bitterman[7], Grace Erne[7], Kristen Machado-Diaz[7], Feriel Presswalla[7], Namratha Nelapudi[7], Kingsley-Michael Amadi[7], Alina Bardwell[7], Pennelope Blakely[7], Yiyuan Huang[7], Mousumi Banerjee[7], Rodica Pop-Busui[7] & Salim S. Hayek [7] ✉

