## [Peer Review File · Nature Communications]

Rapid changes in kidney filtration barrier function are a result of suPAR mediated viral response in COVID-19REVIEWER COMMENTS

Reviewer #1 (Remarks to the Author):

Wei et al have studied non-omicron and omicron virus effects on African Green monkeys, mice, podocytes and humans. They find a clear association between the non-omicron virus to proteinuria which is demonstrated by vertical and mechanistic studies. Overall, a tremendous tour-de-force. Work is novel given that the glomerular filtration barrier has not been studied well during COVID, and very clinically relevant given that we are still having waves of COVID in North America and worldwide.

1. It is unclear why the IL-5 Tg mouse is used and how this affects interpretation of the work.
2. What percent of the Omicron patients had vaccination, which is not clear in Table 2. The vaccinated mice seemed to have less proteinuria.
3. The HIV pathway had similarities to COVID and were distinct from Hep C and Influenza. Could the authors discuss potential pathways that explain these similarities and differences?

Reviewer #2 (Remarks to the Author):

In the manuscript by Wei et al entitled "Virus-response proteinuria underlies rapid changes in kidney filtration barrier function in COVID-19" the authors investigate the role of suPAR in the development of proteinuria in non-humans primates, mice as well as a human cohort infected with COVID-19. Their main findings are:

1. African green monkeys infected with COVID-19 had increased levels of suPAR and proteinuria. They also had disorganized staining patterns of podocin and α -actinin-4 as well reduced filtration slit density. In addition these monkeys had evidence of glomerular injury.
2. In suPAR-Tg mice the inhalation of COVID-19 spike S1 protein had increased proteinuria, foot processes effacement and reduced filtration slit density
3. In these mice the development of proteinuria was reduced by either COVID-19 vaccination or suPAR antibodies
4. Inoculation of Delta S1 protein but not Omicron S1 protein induced proteinuria in suPAR-Tg mice
5. In vitro co-immunoprecipitation assays showed interaction between suPAR and COVID-19 spike protein but no tight binding as assessed by SPR. There was however good binding to α v β 3 integrin
6. Omicron S1 protein showed less tight binding to ACE2 or α v β 3 integrin as compared to non-Omicron S1 protein
7. suPAR increased podocyte ACE2 and α v β 3 expression in podocytes in response to COVID S1 protein but not in response to Omicron S1
8. In a large human cohort patients infected with non-Omicron variants had more proteinuria and lower suPAR levels as compared to those infected with Omicron

Major Comments:

1. These are interesting studies that provide some insight into the pathogenesis of renal disease and proteinuria in COVID-19 infections. The authors provide data derived from African green monkeys, mice and a human cohort although the bulk of the data comes from the mouse experiments.
2. Although the authors provide several lines of evidence for renal injury in monkeys and mice no renal function data is reported which may add support to their findings (i.e creatinine, NGAL, KIM-1, etc). The addition of this data would help to improve the manuscript
3. The mouse data is positive in the suPAR-Tg mice but not in C57 mice. It would help to strengthen their conclusions to have evidence of proteinuria without the need of having increased suPAR expression. The investigators could perhaps test whether COVID-19 infection leads to proteinuria in another mouse strain more susceptible to renal injury or in a rat model that maybe more likely to develop proteinuria as compared to C57 mice.
4. Was there any evidence of long-term renal injury in the different groups studied? Was complete resolution of renal injury after infection was cleared?
5. Given the emphasis presented on the differences based on the type of COVID variant it would be

ideal to have data in primates using the different COVID variants even if done in a low number of animals

Response to referees:

Referee 1:

We thank the referee for the positive comments. Responses to the specific questions are as follows:

1. It is unclear why the IL-5 Tg mouse is used and how this affects interpretation of the work.

Response: In addition to the elevated suPAR levels (Azam et al. J Am Soc Nephrol 2020), an increase in multiple type 2 immune effectors in blood circulation, including IL-5 was often observed in severe Covid-19 patients as well (Lucas et al. Nature 2020; 584:463-469). To get a clue if spike S1 protein synergizes specifically with high level of suPAR to cause proteinuria, we tested IL-5 for control. The selection of IL-5-Tg mouse model was based on not only the relevance of IL-5 in severe COVID-19 patients, but also the following considerations: 1) Given the timely relevance of our findings in the pandemic/immediate post-pandemic time, we chose this model that was readily available in the laboratory; 2) It had high levels of circulating IL-5 and but no overt proteinuria at baseline, a feature similar to suPAR-Tg model. Our findings that Spike S1 protein inoculation induced proteinuria in suPAR-Tg mice but not in IL-5-Tg mice suggest a non-random synergy of S1 protein and suPAR in causing proteinuria.

2. What percent of the Omicron patients had vaccination, which is not clear in Table 2. The vaccinated mice seemed to have less proteinuria.

Response: In the M²C² cohort, 3.7% of non-omicron and 54.6% of Omicron-infected patients had been vaccinated prior to their hospitalization, respectively. Omicron-infected patients that were vaccinated had a lower incidence of 2+ proteinuria compared to non-vaccinated patients (24.0% vs. 19.5%, P=0.15). Albeit supportive of the experimental model findings, the difference did not reach statistical significance, and should be considered as preliminary evidence. The disparity between human patients and mouse models possibly results from the following aspects: 1) Our mouse model is spike S1 induced that could be completely neutralized by vaccine generated anti-spike antibody; 2) Our mouse model, unlike patients does not have preexisting renal disease and/or co-morbidities; 3) Some patients had already contracted SARS-Cov-2 infection before receiving vaccines.

3. The HIV pathway had similarities to COVID and were distinct from Hep C and Influenza. Could the authors discuss potential pathways that explain these similarities and differences?

Response: Thanks for this comment. We have added discussion to the manuscript. HIV and COVID-19 share similarities especially in terms of renal diseases (Cohen et al. N Eng J Med. 2017; Legrand et al. Nat Rev Nephrol. 2021; Chen et al. Kidney Dis. 2022);). Both are associated with a wide range of kidney disease manifestations. Thus,

comparisons of patients and kidneys infected by SARS-CoV-2 and other viral infections, especially HIV infection are justified. Both SARS-CoV-2 and HIV can directly infect podocytes and tubular epithelial cells. Collapsing glomerulopathy, tubulointerstitial fibrosis, and mesangial injury can be seen in both COVID-19 and HIV infection, and are perhaps dependent on the amount of the innate immune activation that is mounted as an anti-infection response (Reiser et al. Cell Metab. 2022). Mechanistically, several considerations are possible. For example, African Americans develop COVID-19 associated with stronger acute kidney injury (AKI) relatively more often than observed in other racial and ethnic groups (Hung et al. JAMA Intern Med. 2022). Data from patients with COVID-19 also suggested AKI, proteinuria, and podocytopathy were more prevalent among those with the high-risk *APOL1* genotype. Notably, suPAR can cooperate with *APOL1* risk proteins to enhance podocyte integrin activation (Hayek et al. Nat Med. 2017), a mechanism that may also be directly or indirectly at play in COVID-19. Given our recent findings, a stage is set for these topics to be investigated further.

Neither influenza nor hepatitis C virus (HCV) has been shown directly to cause renal injuries. Influenza is largely confined to upper respiratory tract, with lower respiratory tract affected in severe cases (Flerlage et al. Nat Rev Microbiol. 2021), while the viral replication of HCV is primarily confined to the liver. For cell attachment and/or entry, the haemagglutinin (HA) protein of influenza viruses preferably binds sialosaccharides on the surface of pulmonary epithelial cells. Uptake of HCV into hepatocytes involves virion-associated lipoproteins, numerous host factors and cell-associated factors (Ding et al. Cell Host & Microbe. 2014). It is not yet clear if any integrin is implicated in either HCV or Influenza infection. HCV could evade immune elimination, leading to chronic infection and accumulation of circulating immune complexes (Bruggeman. Adv Chronic Kidney Dis. 2019; Lai et al. Nat Clin Pract Nephrol. 2019), which may contribute to the development of HCV-associated glomerulonephritis.

Referee 2:

Major Comments:

1. *These are interesting studies that provide some insight into the pathogenesis of renal disease and proteinuria in COVID-19 infections. The authors provide data derived from African green monkeys, mice and a human cohort although the bulk of the data comes from the mouse experiments.*

Response: We thank the referee for the positive comments.

2. *Although the authors provide several lines of evidence for renal injury in monkeys and mice no renal function data is reported which may add support to their findings (i.e creatinine, NGAL, KIM-1, etc). The addition of this data would help to improve the manuscript.*

Response: Upon the reviewer's suggestion, we measured kidney injury biomarkers in the urine samples collected from 2019-nCov inoculated African Green Monkeys

(AGMs). As shown below in **Fig RB1.**, urinary levels of calbindin, and VEGF in 2019-nCov inoculated AGMs were significantly increased as compared to that in controls. There was also a clear trend of increase for KIM-1. These findings further indicate that 2019-nCov inoculation caused kidney injuries in AGM.

Fig RB1. Urinary kidney injury biomarkers in AGMs. * $p < 0.05$. Two-tailed T test.

In terms of spike S1 mouse models, we measured blood creatinine and BUN levels, but did not find any significant changes in any of the tested mouse strains (WT/B6, uPARKO/*Plaur*^{-/-}, suPAR overexpressing/suPAR-Tg), suggesting that there were no overt associated AKI (acute kidney injuries) (**Fig RB2**). We also examined the urinary levels of KIM-1, but did not observe any significant changes in KIM-1 after S1 protein inoculation (**Fig RB3a**). Next, we performed immunofluorescent staining for KIM-1 with frozen kidney sections. While we could appreciate clear increase of KIM-1 expression in some tubules from the suPAR-Tg mice inoculated with original spike S1 (2019-nCov), we did not observe the increase in KIM-1 expression with B6 or uPARKO mice. Comparatively, much less of KIM-1 expression could be appreciated in tubules from the suPAR-Tg mice inoculated with Omicron S1 protein (**Fig RB3b**). Taken together, we present additional data showing that original spike S1 (2019-nCov) causes more severe kidney injuries than its Omicron counterpart in suPAR-Tg mice.

Fig RB2. Inoculation of 2019-nCov S1 protein did not cause AKI in mice. No significant difference was found between any groups ($p > 0.05$). Two-way ANOVA.

Fig RB3. Urinary KIM-1 levels. No significant changes ($p > 0.05$) were observed in any tested mouse strains with either 2019-nCov or Omicron S1 inoculation. Two-way ANOVA. **(A). Tubular KIM-1 expression as indicated by immunofluorescence staining.** KIM-1 expression was particularly increased in suPAR-Tg mice that received 2019-nCov, but not Omicron S1 proteins **(B)**.

3. The mouse data is positive in the suPAR-Tg mice but not in C57 mice. It would help to strengthen their conclusions to have evidence of proteinuria without the need of having increased suPAR expression. The investigators could perhaps test whether COVID-19 infection leads to proteinuria in another mouse strain more susceptible to renal injury or in a rat model that maybe more likely to develop proteinuria as compared to C57 mice.

Response: Upon the referee's suggestion, we tested the renal effect of 2019-nCov S1 protein inoculation in Balb/c mice that are susceptible to Adriamycin (ADR) induced kidney damage. As the regularly used ADR dose causes severe proteinuria and FSGS-like nephropathy in male Balb/c mice, we utilized half dose of ADR (5 mg/kg body weight) to create a preexisting renal injury condition, but not a severe glomerulopathy phenotype. In brief, all experimental Balb/c mice (6 weeks old) were administered with ADR (5 mg/kg body weight, tail vein injection) four days before initiation of 2019-nCov spike S1 protein ($n=6$) or vehicle control ($n=6$) nasal inhalation. Urine was collected respectively before ADR injection (day -3), before (day 0) and after 10 day's (day 11) S1 or vehicle inhalation (**Fig RB4A**). Interestingly, no over proteinuria was observed in any ADR pretreated Balb/c mice after 2019-nCov inoculation (**Fig RB4B**). Serum suPAR levels did not increase after the treatment at day 11 (**Fig RB4C**). Taken together, these data further suggest that synergizing with high levels of suPAR is required for 2019-nCov spike S1 protein to cause proteinuria in mice. Our findings from S1 protein mouse models are in line with those observed from moderate to severe COVID-19 patients in that suPAR levels are correlated with the severity of proteinuria.

Fig RB4. 2019-nCov S1 protein did not cause proteinuria in ADR pretreated Balb/c mice. **A.** Experimental design and treatment schemes. ADR, Adriamycin. I/N, intranasal. q.d, daily. **B.** Proteinuria profiles. **C.** Serum suPAR levels. No significant difference was found between any groups ($p > 0.05$) for **B** and **C**. Two-way ANOVA.

4. Was there any evidence of long-term renal injury in the different groups studied? Was complete resolution of renal injury after infection was cleared?

Response: To determine the long-term renal effect of 2019-nCov S1 protein inoculation, we performed a 6-week longitudinal study in mice with high levels of suPAR. As shown in **Fig RB5**, proteinuria was increased over the course of 10-day's S1 administration. One week after last dose of S1, proteinuria decreased sharply, but still higher than baseline levels. Over next 3 weeks or so, urinary protein levels gradually returned to normal. These results suggest that once SARS-Cov-2 viral infection is controlled, or viral particles stop propagation, S1 protein induced proteinuria will resolve over time.

Fig RB5. Longitudinal study of 2019-nCov S1 induced proteinuria in suPAR-Tg mice.

5. Given the emphasis presented on the differences based on the type of COVID variant it would be ideal to have data in primates using the different COVID variants even if done in a low number of animals.

Response: We appreciate the reviewer's comment. Due to the negative impact of Hurricane IDA to our Non-human Primate facility in Covington, LA, we could not initiate any new experiments with AGMs in any timely manner. That said, reduced clinical signs and symptoms have been already reported with the Omicron variant in monkeys (rhesus macaques) (Munster et al. Sci Adv. 2021; van Doremalen et al. Sci Adv. 2022). Omicron caused attenuated disease in mice and hamster as well (Halfmann et al. Nature. 2022). Attenuated fusogenicity and reduced efficiency in its use of transmembrane serine protease 2 (TMPRSS2) were proposed to account for Omicron's reduced pathogenicity (Suzuki et al. Nature. 2022; Shuai et al. Nature. 2022).

REVIEWERS' COMMENTS

Reviewer #1 (Remarks to the Author):

The authors have addressed the comments and the manuscript is much improved.

Reviewer #2 (Remarks to the Author):

The authors have made a good effort to revise the paper that in general is supportive of the findings described in the original version.

Response to reviewers:

Reviewer #1 (Remarks to the author)

The authors have addressed the comments and the manuscript is much improved.

Response: We thank the reviewer for reviewing our manuscript and supporting its publication.

Reviewer #2 (Remarks to the author)

The authors have made a good effort to revise the paper that in general is supportive of the findings described in the original version.

Response: We thank the reviewer for reviewing our manuscript and supporting its publication.